# Levels of Knowledge, Beliefs, and Practices Regarding Osteoporosis and the Associations with Bone Mineral Density among Populations More Than 40 Years Old in Malaysia

**DOI:** 10.3390/ijerph16214115

**Published:** 2019-10-25

**Authors:** Chin Yi Chan, Shaanthana Subramaniam, Kok-Yong Chin, Soelaiman Ima-Nirwana, Norliza Muhammad, Ahmad Fairus, Pei Yuen Ng, Nor Aini Jamil, Noorazah Abd Aziz, Norazlina Mohamed

**Affiliations:** 1Department of Pharmacology, Universiti Kebangsaan Malaysia Medical Centre, Cheras 56000, Malaysia; chanchinyi94@gmail.com (C.Y.C.); shaanthana_bks@hotmail.com (S.S.); chinkokyong@ppukm.ukm.edu.my (K.-Y.C.); imasoel@ppukm.ukm.edu.my (S.I.-N.); norliza_ssp@ppukm.ukm.edu.my (N.M.); 2Department of Anatomy, Universiti Kebangsaan Malaysia Medical Centre, Cheras 56000, Malaysia; apai.kie@gmail.com; 3Faculty of Pharmacy, Universiti Kebangsaan Malaysia Kuala Lumpur Campus, Jalan Raja Muda Abdul Aziz, Kuala Lumpur 50300, Malaysia; pyng@ukm.edu.my; 4Faculty of Health Science, Universiti Kebangsaan Malaysia Kuala Lumpur Campus, Jalan Raja Muda Abdul Aziz, Kuala Lumpur 50300, Malaysia; ainijamil@ukm.edu.my; 5Department of Family Medicine, Universiti Kebangsaan Malaysia Medical Centre, Cheras 56000, Malaysia; azah@ppukm.ukm.edu.my

**Keywords:** knowledge, awareness, attitude, perception, osteoprotective practices, exercise, calcium supplement, bone mineral density, dual-energy X-ray absorptiometry

## Abstract

Osteoporosis is a skeletal disorder commonly found among the elderly, in which the bones become weak, brittle, and more susceptible to fracture. Adequate knowledge and positive attitude towards the disease and osteoprotective activities may prevent osteoporosis, but comprehensive studies to verify this hypothesis are limited in Malaysia. This study aims to bridge the research gap by determining the levels of knowledge, beliefs, and practices regarding osteoporosis and their associations with bone mineral density (BMD) among men and women ≥ 40 years in Klang Valley, Malaysia. In this cross-sectional study, 786 Malaysians (382 men, 404 women) completed a questionnaire on knowledge, beliefs, and osteoprotective practices, and underwent BMD scan using a dual-energy X-ray absorptiometry device. The current study found moderate levels of knowledge and beliefs regarding osteoporosis but poor osteoprotective practices. Osteoporosis knowledge, beliefs, and practices were significantly different based on subjects’ demographic characteristics (*p* < 0.05). Osteoporosis knowledge and beliefs were correlated significantly with osteoprotective practices (*p* < 0.05). Bone health status of the subjects was associated positively with calcium supplement intake, and negatively with exercise barriers and smoking status of the subjects (*p* < 0.05). However, no significant correlation was noted between osteoporosis knowledge and bone health (*p* > 0.05). Conclusively, despite some correlations between individual components, the detachment between bone health knowledge and beliefs, and osteoprotective practices among Malaysians is apparent. Integrating all three components into a comprehensive osteoporosis prevention program is warranted.

## 1. Introduction

Osteoporosis is a systemic skeletal disorder characterized by microstructural deterioration, impaired bone strength, and an increased propensity to fractures. Osteoporosis occurs when the rate of bone resorption by osteoclasts exceeds the rate of bone formation by osteoblasts [1,2]. Osteoclast differentiation is mediated by receptor activator of nuclear factor-kB ligand [3], which in turn, is influenced by cytokine levels [4]. For example, interleukin (IL)-6, IL-17, tumor necrosis factor-α, and interferon-γ promote bone loss by favoring osteoclast production and inhibiting osteoblast differentiation [5,6]. On the other hand, IL-4, IL-12, and IL-33 are strong suppressors of osteoclast differentiation and they inhibit bone loss [7,8].

As life expectancy continues to increase through demographic transition, osteoporosis is becoming a major global health issue with tremendous clinical, economic, and social impacts [9]. In Asia, the population aged over 50 years old is estimated to increase by 144% from 2013 to 2050 [10]. Concurrently, there is an increase in hip fracture incidence in Asian countries undergoing rapid urbanization, whereby the incidence in highly urbanized countries like Hong Kong and Singapore is similar to developed countries like United States [10]. In developing countries like Malaysia, the direct cost for treatment of hip fracture is estimated to increase from 35.3 million USD in 2018 to 125.4 million USD in 2050 [11]. This increase will cause a significant healthcare burden to the patients because the median monthly household income among Malaysians is only RM 4585 [12]. Given the high cost associated with fragility fracture, early detection and prevention of patients at risk for osteoporosis are critical.

The pharmacological therapy commonly used to treat postmenopausal osteoporosis can be categorized as either antiresorptive or anabolic medications. Antiresorptive medications act to decrease bone resorption, while anabolic medications increase bone formation [13]. Examples of antiresorptive medications include bisphosphonates, selective estrogen receptor modulators, and denosumab [14]. Teriparatide is the only anabolic medication currently available in the market [13]. For prevention of osteoporosis, adequate calcium intake (1000 mg/day) for adults and weight-bearing physical activities are recommended [15,16].

Osteoporosis is preventable, yet the misconception of it as a geriatric disease is prevalent [17]. Although bone health knowledge and osteoprotective lifestyle are essential in osteoporosis prevention [18], their inadequacy has been reported worldwide [19]. Many individuals are unaware of their risk of developing osteoporosis and are not engaging in osteoprotective behaviors regularly [20]. Several studies evaluating knowledge, beliefs, and practices regarding osteoporosis have been conducted In Malaysia [21,22,23,24,25,26,27]. Most of the populations surveyed are undergraduate students [21,22,27] and young adults [23,26]. Only a few studies were carried out among middle-aged and elderly populations at risk for osteoporosis [24,25]. Besides, limited studies explored the ethnic differences in knowledge, beliefs, and practices regarding osteoporosis in Malaysia [21,24,27], and fewer still correlated them with bone health status of the surveyees. Previously, the authors had conducted a pilot study among middle-aged and elderly Malaysian Chinese to address this issue. In the study, a moderate knowledge level and positive health beliefs regarding osteoporosis were noted, but these did not translate to good osteoprotective practices [28]. The current study is an expansion of the pilot study, in which two more ethnic groups were recruited to provide a more comprehensive view of knowledge, beliefs, and practices regarding osteoporosis among Malaysians.

This study aimed to evaluate the association between knowledge, beliefs, and practices regarding osteoporosis and bone health among middle-aged and elderly Malaysians living in Klang Valley, Malaysia. This study is unique because it involved both sexes and three major ethnic groups of Malaysia. To the best of our knowledge, this is the first cross-sectional study that reported the association between knowledge, beliefs, and practices regarding osteoporosis among Malaysians middle-aged and elderly populations and their bone health status. These data could serve as the baseline in formulating an osteoporosis prevention program in Malaysia.

## 2. Subjects and Methods

### 2.1. Study Design

The protocol of this study was reviewed and approved by the Universiti Kebangsaan Malaysia Ethics Committee (approval code: UKM PPI/111/8/JEP 2017-721). This cross-sectional study was conducted among population aged ≥40 years residing in Klang Valley, Malaysia who has not been diagnosed with osteoporosis. Recruitment of subjects was conducted using quota sampling technique from April 2018 to April 2019. An invitation with specific inclusion and exclusion criteria was sent to community centers in Klang Valley and advertised in local newspapers and radio stations. Potential participants were interviewed over the phone to ensure their eligibility. Only subjects fulfilling the inclusion criteria were invited to attend the screening session. The inclusion and exclusion criteria of the subjects were similar to our pilot study [28]. Subjects previously diagnosed with osteoporosis, metabolic bone diseases (Paget’s disease, osteogenesis imperfecta, osteomalacia, and rickets), hypo/hyperparathyroidism, hypo/hypercalcemia, hypo/hyperthyroidism, and/or who were receiving pharmacological treatment for osteoporosis (bisphosphonates, teriparatide, denosumab, and strontium ranelate) or other treatments that could significantly impact bone metabolism (hormone-replacement therapy, sex hormone deprivation therapy, thiazide diuretics, anticonvulsants, antidepressants, glucocorticoids, and thyroid supplements) were excluded. Those having mobility problems, needing a walking aid, having metal implants at the calcaneus, hip, spine, or femoral neck, suffered a lower limb fracture 2 years prior to the screening date, or a low impact fracture after the age of 50 years, or those who could not complete the questionnaire or screening procedure were excluded as well.

The sample size of this study is calculated based on the reported prevalence of suboptimal bone health (osteoporosis and osteopenia) in the Malaysian populations [29] using the formula n = (Z_0.95_)^2^ P[(1−P)/D^2^], whereby Z_0.95_ = 1.96 for a confidence interval of 95%, P = prevalence of osteoporosis, and D = absolute precision of 5%. Chin et al. (2015) reported that the prevalence of suboptimal bone health among Malaysian men and women aged 50 years or above in Klang Valley was 40.6 % and 43.4%. Therefore, the minimal sample size for men and women in this study was 371 and 378 [29].

### 2.2. Subjects

Eligible subjects were invited to a face-to-face interview session. A total of 786 subjects, consisting of 342 Malays, 363 Chinese, 81 Indians or from other ethnic groups, aged ≥ 40 years were recruited. During the interview session, they were informed of the project details, and written consent was obtained before participation. They answered a questionnaire about their demographic details, knowledge, beliefs, and practices regarding osteoporosis. An appointment for body anthropometry and bone health assessment was scheduled after completing the interview.

Subjects’ age was determined from the records on their identification card. Ethnicity, sex, menstrual status, age of menarche, age of menopause, and parity and presence of pre-existing medical conditions and medical treatments were self-declared. Subjects were classified into their respective age clusters; that is, “middle-aged” for those aged 40–59 years [30,31] and “elderly” for those aged ≥60 years [32]. They were also categorized into manual (walking, standing, or carrying heavy objects most of the time) or sedentary workers (sitting at the workplace or vehicle most of the time) based on the nature of their job. Sedentary workers also included unemployed subjects, retirees, and housewives. Based on the current Malaysian census data, the subjects were also classified based on household income into the bottom 40% (B40, with household income <RM 7640), the middle 40% (M40, with household income between RM 7640 and 15,159), and the top 20% (T20, with household income >RM 15,160). Previous studies suggested that socioeconomic status might influence bone health because it was associated with health behaviors, such as smoking, physical activity, and alcohol intake, which were predictors of osteoporosis [33]. Besides, income status was also linked with knowledge regarding osteoporosis [20,23].

### 2.3. Knowledge and Health Beliefs Regarding Osteoporosis

Subjects’ knowledge and health beliefs regarding osteoporosis were tested using a modified Osteoporosis Prevention and Awareness Tool (OPAAT) [34] and Osteoporosis Health Belief Scale (OHBS) (Kim et al., 2013), respectively. The researchers involved in this study discussed and decided the items to be included in both questionnaires. The knowledge questionnaire consisted of six items regarding the general knowledge of osteoporosis and six items on osteoporosis prevention. The subjects answered “true”, “false”, and “don’t know” for each item. A correct answer was given 1 point, and an incorrect answer or “don’t know” was given 0 point. The health beliefs questionnaire covered subjects’ perceived susceptibility and seriousness to osteoporosis (items 1–3), benefits of exercise and calcium (items 4–5), barriers to exercise and calcium intake (items 6–9), and health motivation (items 10–12). Each item was scored with a Likert’s scale from 1 (strongly agree) to 5 (strongly disagree). The scores of negatively worded items were inversely coded by the researchers during analysis. The scores of knowledge or health belief questionnaire were totaled up, and subjects were categorized into low (less than or equal to 50%), moderate (51–69%), and high (70% and above) knowledge/belief levels [22,23]. The reliability coefficient obtained through test–retest in the pilot study involving 30 subjects with similar characteristics of the current study (seven days apart between the first and second test) for the knowledge questionnaire was 0.739, while that obtained for the health beliefs questionnaire was 0.731.

### 2.4. Dietary and Lifestyle Practices

In terms of practices, the subjects were requested to disclose their smoking behavior, intake of beverage, and intake of dairy product. Subjects answered “yes/no” for the consumption of these products for the past seven days. If they answered “yes”, the type of products and the frequency of consumption (how many times per week) were asked. The current and former smokers (ceased smoking ≥12 months) were combined as “ever-smokers”. The beverages and dairy products investigated included (1) coffee or tea; (2) alcohol (beer, wine, or spirits); (3) milk; (4) yoghurt or cheese. One unit of milk was defined as 200 mL, whereas coffee/tea was defined as one standard coffee cup/tea cup. One unit of yoghurt and cheese was defined as a cup of yoghurt and a slice of cheese, respectively. One alcohol unit was defined according to the recommendation by the National Health Service, UK [35]. A unit of alcoholic drink referred to a bottle of beer/cider, a glass of wine, or one portion of spirits/strong alcohol [36]. For beverages, subjects with an intake of less than one unit per week were defined as non-drinker. Regular coffee/tea drinkers and dairy products consumers were defined as those who consume one unit of coffee/tea or dairy products for three to four days per week. Subjects who consumed alcohol regularly (three to four days per week) or had stopped drinking for ≥12 months were combined as “ever-drinkers”. Those who never or rarely consume alcohol (one or fewer days per month) were combined as “non-drinkers”. Those who consumed at least one tablet of calcium supplement for three to four days per week were considered as regular users.

### 2.5. Physical Activity Assessment

Physical activity status of the subjects was determined using the International Physical Activity Questionnaire (IPAQ)-short form, which is available online and free for use [37]. Briefly, the average amount of time spent in high- and moderate-intensity activity, walking, and sitting/lying down (except sleeping) in a week was recalled by the subjects. The time and frequency of each type of activity were converted to the metabolic equivalent of task (MET) and summed up. Subjects were classified into inactive, minimally active, or HEPA (health-enhancing physical activity) based on the total MET score or other additional criteria [38]. The validity and reliability of this instrument have been tested in the Malaysian population [37].

### 2.6. Body Anthropometry Measurements

Anthropometric measurements were collected by trained research assistants. Height was measured to the nearest 1 cm using a stadiometer (Seca, Hamburg, Germany). Body weight of the subjects with light clothing and without shoes was measured to the nearest 0.1 kg using a weighing scale (Tanita, Tokyo, Japan). Body mass index (BMI) of the subjects was calculated by the formula: body weight in kg divided by the square of height in meter. Generally, for subjects ≤ 65 years old, BMI < 18.5 kg/m^2^ were underweight, 18.5–24.9 kg/m^2^ were normal, 25.0–29.9 kg/m^2^ were overweight, and >30.0 kg/m^2^ were obese [39]. For people > 65 years old, BMI between 22 and 27 kg/m^2^ were normal, >27 kg/m^2^ were overweight, and <22 kg/m^2^ were underweight [40]. The waistline of the subjects was measured to the nearest 0.1 cm by using a soft measuring tape. The waist circumference was measured between the lowest rib margin and the iliac crest while subjects maintained a standing position.

### 2.7. Bone Mineral Density Assessment

Bone mineral density (BMD) of the subjects was determined using a Hologic Discovery QDR Wi densitometer, DXA (Hologic, MA, USA). The DXA machine was calibrated daily with a phantom. Full body, lumbar spine (L1–L4), and left hip scans of each subjects were performed and analyzed upon test completion, with each scan taking approximately 5 min. The short-term in vivo coefficient of variation for the DXA machine was 1.8% for the lumbar spine and 1.2% for the total hip. The body fat percentage, lean body mass, and lumbar and hip BMD were computed automatically by the DXA scanner. The T-score was generated by comparing the BMD values of the subjects with the reference values of the Singaporean population because local reference is not available. According to the guidelines of World Health Organization, a T-score of ≤−2.5 indicates osteoporosis, between −2.5 and −1 indicates osteopenia, and >−1 indicates normal bone health status [41].

### 2.8. Statistical Analysis

Statistical analysis was performed using Statistical Package for Social Science Version 22 (IBM, Armonk, NY, USA). Statistical significance was set as *p* < 0.05. All variables were assessed for normality using Kolmogorov–Smirnov test. Continuous variables were expressed as mean (standard deviation), while categorical variables were expressed as count and percentage. Independent T-test or one-way analysis of variance with Tukey’s or Dunnet T3 post hoc analysis was used to determine the difference in sociodemographic characteristics, knowledge and belief level related to bone health. Chi-square analysis was used to determine difference in sociodemographic characteristics and practices related to bone health. Pearson correlation was used to identify the relationship among knowledge, beliefs, and practices with subjects’ bone health status.

## 3. Results

### 3.1. Characteristics of Subjects

A total of 786 subjects, comprising 382 men and 404 women, were enrolled in the study. The men were significantly older, taller, heavier, had lower fat percentage, higher lean mass, and wider waist circumference as compared with the women (*p* ≤ 0.001). Among the 404 women recruited, 265 were menopausal women, with an average of 9.68 (SD = 6.68) years since menopause. According to ethnicity, 46.2% of the subjects were Chinese, 43.5% were Malays, and 10.3% were Indian or others. Most of the subjects were from Hulu Langat district (79.5%), married (93.4%), having a sedentary job (94.1%), and from B40 income group (93.1%). The majority had secondary school education (47.5%). In terms of dietary and lifestyle practice, 62.6% of the subjects rarely consumed dairy products, and 85.0% did not take a calcium supplement regularly. Most of the subjects drank coffee and tea regularly (87.8%), but the majority were non-smokers (77.7%) and non-alcohol drinkers (87.8%). Only 12% of the subjects were active in physical activity (HEPA-active). With regards to bone health, 49.6% of the subjects had normal BMD, 38.0% had osteopenia, and 12.3% had osteoporosis. Characteristics of the study population is shown in Table 1. 

### 3.2. Knowledge and Belief Toward Osteoporosis

Overall, the mean total knowledge score was 68.64% (SD = 12.83). Subjects scored higher in general knowledge (72.67%, SD = 17.00%) than prevention knowledge regarding osteoporosis (64.61%, SD = 16.45%). Most of the subjects correctly identified the phrase “osteoporosis makes bones weaker, brittle, and more likely to break, causing fractures” as being “true” (95.8%). Additionally, they were aware of the importance of having good vision and comfortable shoes with good grip to prevent them from falling (93.8%). They were also conscious about dietary sources rich in calcium such as milk, tofu, anchovies, yellow dhal (lentils), and spinach (93.3%). However, most of the subjects could not correctly identify the phrase “the regular intake of calcium supplements can lead to the formation of kidney stones” (13.7%) and “osteoporosis will result in knee pain” as being “false” (21.1%). Descriptive analysis for each item in the questionnaire, in general and based on sexes can be found in Appendix A.

For health beliefs regarding osteoporosis, the average total score was 63.57% (SD = 5.66). Subjects perceived moderate susceptibility towards osteoporosis (57.88%, SD = 14.11). The perception of subjects on the seriousness of osteoporosis, benefits of exercise, and benefits of calcium intake was high. Subjects also perceived that they had low barriers to exercise and calcium intake. For calcium intake, 86.7% of the subjects agreed that they liked calcium-rich foods, and 76.1% agreed that calcium food was not costly. For exercise, 65.4% of them felt that they are strong enough to exercise regularly, while 64.3% of them stated that starting a new habit to exercise regularly is not hard. High health motivation was shown by subjects, whereby they were willing to look for new information related to health (75.8%), do regular health check-up (66.7%), and follow the recommendation to keep them healthy (90.5%). Descriptive analysis for each item in the questionnaire, in general and based on sexes can be found in Appendix A. The level of knowledge and beliefs regarding bone health of the subjects is shown in Table 2.

### 3.3. Comparison of Sociodemographic Factors on Knowledge and Belief Regarding Osteoporosis

Knowledge regarding osteoporosis was associated with sex, ethnicity, monthly income, and education level. General knowledge regarding osteoporosis of women (75.91%) was significantly higher compared with men (69.24%) (*p* ≤ 0.001). On the other hand, Malay subjects (67.25%) had significantly higher knowledge regarding prevention of osteoporosis compared with Chinese (61.80%), Indian subjects and those from other ethnic groups (66.05%) (*p* ≤ 0.001). Subjects with monthly income >RM 15,160 (T20) (*p* = 0.043) and those with at least a university degree (*p* ≤ 0.001) also had significantly higher general knowledge regarding osteoporosis.

In terms of health beliefs, significant differences were found among age, sex, ethnic, job, education, and menstrual categories. The elderly subjects in this study had significantly lower barriers to exercise (*p* ≤ 0.001) and higher health motivation (*p* = 0.026) compared with the middle-aged subjects. Comparison between sexes showed that women perceived significantly higher susceptibility to osteoporosis (59.65%) compared with men (56.00%) (*p* ≤ 0.001). Women (43.91%) also had lower barriers to calcium intake compared with men (45.80%) (*p* = 0.005). In contrast, men perceived higher benefits of exercise (*p* = 0.014) and lower barriers to exercise (*p* ≤ 0.001) compared with women. Among ethnic groups, Malay had higher perception on seriousness of osteoporosis (*p* = 0.009) and benefits of calcium intake (*p* ≤ 0.001), while Chinese had lower barriers to exercise (*p* ≤ 0.001) and calcium intake compared with others (*p* = 0.017). Subjects with sedentary jobs perceived higher benefits of exercise than manual workers (*p* = 0.018). Those with at least secondary school education perceived higher susceptibility towards osteoporosis (*p* = 0.017), while those with at least a university degree had lesser barriers to exercise (*p* = 0.002) and higher health motivation (*p* ≤ 0.001). Among women, peri-menopausal women perceived significant higher susceptibility to osteoporosis compared with the others (*p* = 0.012), whereas menopausal women had lesser barriers to exercise compared with the others (*p* = 0.036) (Table 3).

### 3.4. Comparison of Osteoprotective Practices with Sociodemographic Factors

Dietary and/or lifestyle practices were significantly different based on age, sex, ethnicity, nature of job, and education level in this study. More elderly subjects (18.2%) were regular calcium supplement users compared with middle-aged subjects (12.7%). More middle-aged subjects (52.3%) were inactive compared with elderly subjects (42.6%). Sex and ethnic differences were revealed in all dietary and lifestyle practices (*p* < 0.05). More women consumed dairy products (45.5% vs. 28.8%) and calcium supplements (19.1% vs. 10.7%) regularly than men. More men consumed coffee or tea (88.6% vs. 76.5% in women) and alcohol (20.4% vs. 4.5% in women), smoked cigarettes (41.6% vs. 3.2% in women) regularly, and were physically active (14.4% vs. 9.7% in women) than women. Among the ethnic groups, most of the regular consumers of dairy products were Indians, while most of the smokers were Malays (*p* < 0.05). Most regular calcium supplement and alcohol users were Chinese. Most physically active subjects were also Chinese compared with other ethnic groups (*p* < 0.05). On the other hand, most manual workers were smokers (34.8% vs. 21.2% among sedentary) and physically active (15.2% vs. 11.8% among sedentary). Additionally, more subjects with primary school education or below consumed alcohol regularly (*p* = 0.031) (Table 4). 

### 3.5. Correlation between Osteoporosis Knowledge, Osteoporosis Health Belief, Osteoprotective Practices, and Bone Health Status

This study demonstrated that knowledge regarding osteoporosis was correlated with several aspects of health beliefs regarding osteoporosis and osteoprotective practices. Positive correlation was noted between osteoporosis knowledge and perceived benefits of exercise (r = 0.169, *p* ≤ 0.001), perceived benefits of calcium intake (r = 0.151, *p* ≤ 0.001), health motivation (r = 0.219, *p* ≤ 0.001), dairy intake (r = 0.108, *p* = 0.002), as well as calcium supplement intake (r = 0.086, *p* = 0.016). On the other hand, a negative relationship was found between osteoporosis knowledge and barriers to exercise (r = −0.094, *p* = 0.008), barriers to calcium intake (r = −0.145, *p* ≤ 0.011), and smoking status (r = −0.083, *p* = 0.019). Health beliefs regarding osteoporosis correlated significantly with several dietary and lifestyle practices of the subjects. Perceived benefits of calcium intake (r = 0.071, *p* = 0.046), low barriers of calcium intake (r = −0.080, *p* = 0.024), and high health motivation (r = 0.164, *p* ≤ 0.001) were associated with high dairy intake. In addition, calcium supplement intake was associated with perceived susceptibility of osteoporosis (r = 0.078, *p* = 0.028) and health motivation (r = 0.073, *p* = 0.040). Additionally, low barrier to exercise was also associated with higher alcohol drinking (r = −0.104, *p* = 0.003). Subjects with higher barriers to calcium intake (r = 0.094, *p* =0.009) and low health motivation (r = −0.072, *p* = 0.044) tended to be smokers. Perceived benefits of exercise (r = 0.110, *p* = 0.002), low barrier to exercise (r = −0.205, *p* ≤ 0.001), and high health motivation (r = 0.120, *p* = 0.001) predicted higher physical activity status. Knowledge regarding osteoporosis was not associated with the bone health status of the subjects (*p* < 0.05). However, several aspects of health beliefs regarding osteoporosis, dietary and lifestyle practices were correlated with bone health status of the subjects. A positive relationship was indicated between calcium supplement intake (r = 0.082, *p* = 0.021) and bone health status of the subjects. A negative relationship between barriers of exercise (r = −0.063, *p* = 0.041) and smoking status (r = −0.079, *p* = 0.026) was also noted (Table 5). 

## 4. Discussion

Osteoporosis knowledge is one of the factors associated with osteoporosis preventive behavior [42]. In the present study, a moderate level of knowledge regarding osteoporosis was indicated among the subjects. This observation was in line with previous studies on knowledge regarding osteoporosis among Malaysian adult populations [23,24]; adult women (aged >40 years old) in Alexandria, Egypt [43], and Arabian women (aged 20 to 44 years old) in Qatar [44]. The study participants were aware of the general and prevention knowledge regarding osteoporosis, that is, it might increase fracture risk, is a treatable disease, and is associated with menopause. Subjects were also aware of the importance of calcium supplements in preventing osteoporosis. The awareness that osteoporosis is preventable may be used as a strategy to stimulate the subjects to accept and comply with the health education messages regarding prevention of osteoporosis. Moreover, subjects’ awareness of the availability of treatments for osteoporosis can motivate them to undergo bone mineral density assessment and treatment [20,45]. However, several misconceptions related to osteoporosis among the subjects must be corrected. Firstly, 64.8% of the subjects were unaware that osteoporosis does not cause knee pain, presumably because the subjects failed to understand the difference between osteoarthritis with osteoporosis. This misconception should be addressed in the future osteoporosis education campaign. Half of the subjects also indicated that regular intake of calcium supplements can lead to the formation of kidney stones, which might become a barrier for them to consume calcium supplements. Additionally, 51.7% subjects did not know that glucocorticoids could increase the risk of osteoporosis, probably because they were not familiar with glucocorticoids and their side effects. Therefore, secondary osteoporosis induced by common medications like glucocorticoids should be included as part of the education program, especially to the patients using these medications.

This study revealed that level of knowledge regarding osteoporosis was different with regards to sex, ethnicity, income, and education level of the subjects. The observation that women were better informed about the general aspects regarding osteoporosis was similar with many previous studies [46,47,48,49]. Women may be more concerned about this disease because they are more susceptible to osteoporosis. Furthermore, Malay subjects were shown to have a significantly higher knowledge level regarding prevention of osteoporosis compared with other ethnic groups. This finding contradicted previous studies among Malaysians adults, whereby most studies reported no ethnic difference in knowledge regarding osteoporosis [23,26]. This might be contributed to the fact that the Malay subjects in this study resided in the urban area, where health information was highly accessible. Similar to previous studies, subjects with higher monthly income (T20) or at least a university degree also had significantly higher general knowledge regarding osteoporosis [23,49,50,51]. Subjects with higher socioeconomic status and better education might have greater access to health information, either through the internet or through healthcare providers. This observation also highlights a challenge for an osteoporosis campaign to reach out to populations with lower socioeconomic background and less education.

This study also highlighted a moderate level of beliefs regarding osteoporosis among the subjects. Significant sex differences were found for perceived susceptibility and benefits of exercise, as well as barriers to exercise and calcium intake. Men’s perceived lower susceptibility to osteoporosis and higher barriers of calcium intake may be because osteoporosis was inaccurately regarded as a “female disease”. Women’s barriers to calcium intake were low, probably because they had a high awareness regarding osteoporosis prevention. Meanwhile, men perceived more benefits and lower barriers to exercise than women, which was similar to findings in other countries [46,52,53]. High barriers of women to exercise may be due to family responsibility (e.g., busy with house chores or taking care of children, lack of single-sex facilities, and safety concerns) [54,55]. However, elderly in this study, especially post-menopausal women, showed lower exercise barriers and higher health motivation compared with the middle-aged subjects. The elderly, who were mostly retirees, might have more spare time to engage in physical activity. They might be also more conscious about taking care of their health due to their age. With regards to ethnic differences in osteoporosis beliefs, the Malays perceived greater seriousness of osteoporosis and benefits of calcium intake. This may be because they lived with seniors with osteoporosis at home, thus were aware that osteoporosis is a crippling disease and understood the importance of calcium intake to prevent osteoporosis. However, the actual reason required further investigation. Meanwhile, the Chinese had lesser barriers to exercise and calcium intake, probably due to the awareness of their higher susceptibility to osteoporosis and the needs for prevention. A similar observation was obtained in a previous study, showing that the elderly Malaysian Chinese were more active than their Indian and Malay counterparts [56]. A significant higher perceived susceptibility towards osteoporosis, higher health motivation, and lesser barriers to exercise discrepancies were found in subjects with higher education level. It was postulated that highly educated subjects were more motivated to exercise because they were aware of the advantages of exercise [57]. Besides, they might have greater access to facilities to exercise [57]. Furthermore, peri-menopausal women’s perceived higher susceptibility towards osteoporosis may be because the symptoms of menopause began to appear at this stage [58]. Subjects with sedentary jobs perceived higher benefits of exercise, probably because they performed physical activity for health purposes rather than as a job requirement.

Dietary (dairy products, calcium supplements, coffee or tea) and lifestyle practices (smoking, alcohol, physical activity) among subjects were also examined in this study. Lifestyle factors play an important role in the pathogenesis of osteoporosis. Despite a moderate level of knowledge and health beliefs regarding osteoporosis, osteoprotective practices among subjects in this study were poor. Other studies also observed similar discrepancies between knowledge, beliefs, and practices regarding osteoporosis [49,59]. Osteoporosis preventive practices were not performed regularly by the subjects, because only 37.4% consumed dairy products and 15.0% consumed calcium supplements. Some of the possible barriers to obtaining adequate calcium intake, as reported by previous studies, were uncertainty regarding calcium food sources and supplements, and concerns related to weight gain and the fat and cholesterol content of some calcium-rich foods [60]. Additionally, in the current study, women and the elderly tended to consume dairy products and calcium supplements regularly, maybe because they felt more susceptible to bone loss. While Indian subjects took dairy products regularly, more Chinese subjects chose to take calcium supplements regularly. This difference could be due to that dairy products are a common ingredient of the Indian diet but not for the Chinese. Therefore, the Chinese choose to compensate by taking calcium supplements. On the other hand, only 12% of the subjects in the current study were physically active, which was incongruent with a previous report that Malaysia is one of the least physically active countries in the world, with over 60% of adults being essentially sedentary [61]. This study further revealed that subjects who were women, middle-aged, Malay, and having a sedentary job were less active compared with others. This might relate to the traditional views of the society on women, whereby physical activity is seen as unfeminine and associated with lower social status [62].

This study also illustrated that knowledge and health beliefs regarding osteoporosis were correlated with osteoprotective practices. It was noted that subjects with higher knowledge and beliefs regarding osteoporosis were more health motivated, aware of the benefits of osteoprotective practices, and have lesser barriers on exercise and calcium intake. They also frequently engaged with osteoprotective practices, for example, taking dairy products and calcium supplements. These observations show that knowledge and positive attitude are essential to motivate individuals to adopt osteoprotective practices. Rationally, health beliefs depend on an individual’s perception of the health problem. However, perception could be modified by knowledge about the disease. If an individual’s perception of the health problem dictates his health behaviors, the improvement in health beliefs is likely to be beneficial in changing the person’s lifestyle [63]. Good knowledge and awareness will lead to lifestyle modifications and the prevention of diseases.

Additionally, the associations between knowledge, beliefs and practices, and bone health of subjects were also determined in this study. No significant correlation was found between subjects’ bone health and osteoporosis knowledge within this study. This finding contradicted the observation obtained among postmenopausal women in Silesia Osteo Active Study, which revealed a positive influence of the knowledge of osteoporosis on femoral neck density in postmenopausal women without prior personal experience of the disease [64]. However, several aspects of health beliefs regarding osteoporosis, dietary and lifestyle practices were correlated with bone health status of the subjects. A positive relationship was indicated between calcium supplement intake and bone health status of the subjects. Calcium supplement is widely used as a pharmacological agent to prevent bone loss [65,66]. A meta-analysis reported that calcium supplements increased bone mineral density of a person at all sites over one to two years (by 0.7–1.8%) [66]. On the other hand, a negative relationship between barriers of exercise, smoking status and bone health status was also noted. It was shown those subjects with lower barriers of exercise have better bone health. Exercise plays an important role in maintaining bone health because it exerts mechanical stress and stimulates the increase in bone mass [67]. In contrast, long-term cigarette smoking could compromise bone density by inducing oxidative stress and inflammation in the body. Both of these processes favor bone resorption and lead to bone loss [68,69].

The present study should be interpreted within the context of its strength and limitations. It is limited by its cross-sectional design, whereby the long-term effects of knowledge, health beliefs, and practices regarding osteoporosis on bone loss were not assessed longitudinally. The current study excluded individuals with high risk of osteoporosis and prior fractures, thus the subjects recruited could be healthier than the general population. The full version of OPAAT and OHBS were not adopted to evaluate the knowledge and beliefs because most subjects could not complete the questionnaires in the pilot study due to attrition. Hence, the researchers selected the most relevant questionnaires and retested them in the pilot study. The daily dietary calcium intake of the subjects was not examined in detailed using a food frequency questionnaire. Despite these limitations, it was the first study that attempted to determine the association between knowledge, beliefs, and practices regarding osteoporosis and bone health status among middle-aged and elderly Malaysians of both sexes and involving all three main ethnic groups in Malaysia using DXA.

## 5. Conclusions

In summary, the present study showed a moderate level of knowledge and beliefs regarding osteoporosis but poor osteoprotective practices among middle-aged and elderly Malaysians. Women had higher general knowledge and perceived susceptibility to osteoporosis, but men had fewer barriers to exercise and higher health motivation. More women consumed calcium and dairy products regularly than men. Knowledge and beliefs regarding osteoporosis were associated with osteoprotective practices in this study. No significant correlation was found between subjects’ bone health determined by DXA and osteoporosis knowledge regarding osteoporosis in this study. However, several aspects of health beliefs regarding osteoporosis, dietary and lifestyle practices were correlated with bone health status of the subjects. The gap between osteoporosis awareness and practices is a major challenge to be addressed in a future osteoporosis prevention campaign. It is suggested that a future awareness campaign should associate positive health behaviors like weight-bearing exercise with osteoporosis prevention as the general population may not be aware that they are linked.

## Figures and Tables

**Table 1 ijerph-16-04115-t001:** Characteristics of the study population.

Variable of Interest	Mean (SD)		
	Men (n = 382)	Women (n = 404)	Overall (n = 786)
**Age (years)**	58.35 (9.41) ^a^	56.03 (8.70)	57.16 (9.12)
Age of menarche (years)	-	13.17 (1.73)	-
Number of children (n)		3.08 (2.13)	
Age of menopause (years)		51.17 (3.45), n = 265	
Years since menopause (years)		9.68 (6.68), n = 265	
**Body Anthropometry**			
Height (m)	166.54 (9.67) ^a^	154.60 (5.50)	160.39 (9.84)
Weight	70.90 (10.78) ^a^	61.08 (12.30)	65.85 (12.56)
BMI (kg/m^2^)	25.42 (3.61)	25.54 (4.98)	25.48 (4.36)
Body fat percentage (%)	29.62 (4.62) ^a^	40.22 (5.35)	35.06 (7.29)
Lean body mass	46.90 (5.92) ^a^	34.00 (5.52)	40.27 (8.61)
Waist circumference (cm)	89.20 (10.68) ^a^	82.81 (11.43)	85.91 (11.52)
	**n (%)**		
**Age Range**			
Middle age (40–59 years old)	195 (51.0)	262 (64.9)	457 (58.1)
Elderly (60 years old and above)	187 (49.0)	142 (35.1)	329 (41.9)
**Ethnicity**			
Malay	160 (41.9)	182 (45.0)	342 (43.5)
Chinese	181 (47.4)	182 (45.0)	363 (46.2)
Indian and others	41 (10.7)	40 (9.9)	81 (10.3)
**District**			
Klang	12 (3.1)	12 (3.0)	24 (3.1)
Kuala Langat	1 (0.3)	1 (0.2)	2 (0.3)
Hulu Langat	291 (76.2)	334 (82.7)	625 (79.5)
Hulu Selangor	2 (0.5)	-	2 (0.3)
Petaling	54 (14.1)	36 (8.9)	90 (11.5)
Gombak	21 (5.5)	19 (4.7)	40 (5.1)
Sepang	1 (0.3)	2 (0.5)	3 (0.4)
**Marital Status**			
Single	15 (3.9)	37 (9.2)	52 (6.6)
Married	367 (96.1)	367 (90.8)	734 (93.4)
**Nature of Job**			
Manual	27 (7.1)	19 (4.7)	46 (5.9)
Sedentary	355 (92.9)	385 (95.3)	740 (94.1)
**Classification of Monthly Incomes**			
B40	344 (90.1)	388 (96.0)	732 (93.1)
M40	36 (9.4)	16 (4.0)	52 (6.6)
T20	2 (0.5)	-	2 (0.3)
**Highest Education Level**			
No formal education	1 (0.3)	5 (1.2)	6 (0.8)
Primary school	30 (7.8)	27 (6.7)	57 (7.3)
Secondary school	160 (41.9)	213 (52.7)	373 (47.5)
Certificate/diploma	93 (24.3)	82 (20.3)	175 (22.3)
University degree	57 (14.9)	53 (13.1)	110 (14.0)
Postgraduate	41 (10.7)	24 (5.9)	65 (8.3)
**Current Menstrual Status**			
Pre-menopause	-	99 (24.5)	-
Peri-menopause		40 (9.9)	
Menopause		265 (65.6)	
**Number of Lifetime Pregnancies (Parity)**			
Nulliparous	-	70 (17.3)	-
1–3 Pregnancies		179 (44.3)	
More than 3 Pregnancies		155 (38.4)	
**Dairy Intake**			
Do not drink	272 (71.2)	220 (54.5)	492 (62.6)
Regular drinker	110 (28.8)	184 (45.5)	294 (37.4)
**Calcium Supplement Intake**			
Yes	41 (10.7)	77 (19.1)	118 (15.0)
No	341 (89.3)	327 (80.9)	668 (85.0)
**Coffee or Tea Intake**			
Do not drink	51 (13.4)	95 (23.5)	146 (18.6)
Regular drinker	331 (86.6)	309 (76.5)	640 (81.4)
**Alcohol Drinking**			
Never	304 (79.6)	386 (95.5)	690 (87.8)
Rarely	25 (6.5)	12 (3.0)	37 (4.7)
Regular drinker	15 (3.9)	-	15 (1.9)
Former drinker	38 (9.9)	6 (1.5)	44 (5.6)
**Smoking Status**			
Never	220 (57.6)	391 (96.8)	611 (77.7)
Current smoker	78 (20.4)	6 (1.5)	84 (10.7)
Former smoker	84 (22.0)	7 (1.7)	91 (11.6)
**Physical Activity Status**			
Inactive	168 (44.0)	211 (52.0)	379 (48.2)
Minimally-Active	159 (41.6)	154 (38.1)	313 (39.8)
HEPA Active	55 (14.4)	39 (9.7)	94 (12.0)
**Body Mass Index**			
Normal	173 (45.3)	177 (43.8)	320 (44.5)
Underweight	26 (6.8)	39 (9.7)	65 (8.3)
Overweight	183 (47.9)	188 (46.5)	371 (47.2)
**Bone Health Status**			
Normal	226 (58.9)	164 (40.8)	390 (49.6)
Osteopenia	124 (32.5)	175 (43.3)	299 (38.0)
Osteoporosis	32 (8.4)	65 (16.1)	97 (12.3)

SD: standard deviation; a: indicates significant difference of *p* < 0.05, as assessed using independent t-test; B40, subjects with household income <RM7640; M40, subjects with household income RM 7640–15,159; T20, subjects with household income >RM 15,160. BMI: body mass index; HEPA: health-enhancing physical activity.

**Table 2 ijerph-16-04115-t002:** Level of knowledge and beliefs towards osteoporosis.

Aspects	Overall (n = 786), n (%)	Mean (%) (SD)
**General Knowledge Regarding Osteoporosis (Q1–6)**		
Low (0–50%)	138 (17.6)	72.67 (17.00)
Moderate (51–69%)	246 (31.3)	
High (70–100%)	402 (51.1)	
**Knowledge Regarding Prevention of Osteoporosis (Q7–12)**		
Low (0–50%)	264 (33.6)	64.61 (16.45)
Moderate (51–69%)	314 (39.9)	
High (70–100%)	208 (26.5)	
**Total Knowledge regarding Osteoporosis (Q1–12)**		
Low (0–50%)	107 (13.6)	68.64 (12.83)
Moderate (51–69%)	310 (39.4)	
High (70–100%)	369 (46.9)	
**I: Perceived Susceptibility to Osteoporosis (Q1-2)**		
Low (0–50%)	323 (41.1)	57.88 (14.11)
Moderate (51–69%)	257 (32.7)	
High (70–100%)	206 (26.2)	
**II: Perceived Seriousness of Osteoporosis (Q3)**		
Low (0–50%)	137 (17.4)	73.28 (18.61)
Moderate (51–69%)	96 (12.2)	
High (70–100%)	553 (70.4)	
**III: Perceived Benefits of Exercise (Q4)**		
Low (0–50%)	35 (4.5)	80.51 (12.88)
Moderate (51–69%)	42 (5.3)	
High (70–100%)	709 (90.2)	
**IV: Perceived Benefits of Calcium Intake (Q5)**		
Low (0–50%)	42 (5.3)	78.37 (12.50)
Moderate (51–69%)	62 (7.9)	
High (70–100%)	682 (86.8)	
**V: Barriers to Exercise (Q6–7)**		
Low (0–50%)	483 (61.5)	50.94 (15.64)
Moderate (51–69%)	154 (19.6)	
High (70–100%)	149 (19.0)	
**VI: Barriers to Calcium Intake (Q8–9)**		
Low (0–50%)	662 (84.2)	44.81 (9.29)
Moderate (51–69%)	94 (12.0)	
High (70–100%)	30 (3.8)	
**VII: Health Motivation (Q10–12)**		
Low (0–50%)	8 (1.0)	74.47 (10.35)
Moderate (51–69%)	240 (30.5)	
High (70–100%)	538 (68.4)	
**Total Belief Regarding Osteoporosis (Q1–12)**		
Low (0–50%)	5 (0.6)	63.57 (5.66)
Moderate (51–69%)	656 (83.5)	
High (70–100%)	125 (5.9)	

**Table 3 ijerph-16-04115-t003:** Comparison of osteoporosis knowledge and beliefs with sociodemographic factors.

Variable	Categories	N	Mean (%) (SD)										
			Knowledge			Health Beliefs							
			General	Prevention	Total	I	II	III	IV	V	VI	VII	I–VII
**Age Range (Years)**	40–59	457	73.23 (16.20)	64.33 (16.17)	68.78 (12.48)	58.56 (14.51)	73.35 (18.72)	80.18 (13.64)	78.69 (12.77)	52.58 (16.29)	44.84 (9.39)	73.77 (10.33)	63.79 (5.76)
	60 and above	329	71.88 (18.05)	65.00 (16.86)	68.44 (13.33)	56.93 (13.50)	73.19 (18.47)	80.97 (11.75)	77.93 (12.12)	48.66 (14.42)	44.77 (9.17)	75.44 (10.33)	63.26 (5.51)
	***p*** **-value**		**0.274**	**0.578**	**0.713**	**0.111**	**0.908**	**0.392**	**0.405**	**≤0.001 ^a^**	**0.924**	**0.026 ^a^**	**0.198**
**Sex**	Men	382	69.24 (18.67)	64.62 (16.67)	66.93 (13.69)	56.00 (14.45)	74.08 (18.88)	81.68 (13.49)	78.53 (12.80)	47.62 (15.02)	45.80 (9.82)	74.26 (10.61)	62.98 (6.02)
	Women	404	75.91 (14.26)	64.60 (16.26)	70.26 (11.76)	59.65 (13.56)	72.52 (18.33)	79.41 (12.19)	78.22 (12.23)	54.08 (15.60)	43.91 (8.69)	74.67 (10.11)	64.12 (5.25)
	***p*** **-value**		**≤0.001 ^a^**	**0.992**	**≤0.001 ^a^**	**≤0.001 ^a^**	**0.241**	**0.014 ^a^**	**0.723**	**≤0.001 ^a^**	**0.005 ^a^**	**0.578**	**0.005 ^a^**
**Ethnicity**	Malay	342	72.22 (16.79)	67.25 (15.78)	69.74 (12.76)	57.22 (13.90)	75.03 (18.38)	80.53 (13.03)	81.11 (10.83)	53.51 (16.39)	45.58 (10.49)	74.39 (11.47)	64.37 (5.87)
	Chinese	363	73.42 (17.15)	61.80 (16.92)	67.61 (13.04)	58.54 (14.41)	72.78 (18.56)	80.50 (12.91)	75.87 (13.60)	48.48 (14.55)	43.80 (8.30)	74.07 (9.39)	62.75 (5.45)
	Indian and others	81	71.19 (17.28)	66.05 (15.24)	68.62 (11.95)	57.65 (13.72)	68.15 (18.92)	80.49 (12.24)	78.02 (11.66)	51.11 (15.57)	46.05 (9.83)	76.63 (9.32)	63.85 (5.21)
	***p*** **-value**		**0.462**	**≤0.001 ^b^**	**0.089**	**0.460**	**0.009 ^b^**	**0.999**	**≤0.001 ^b^**	**≤0.001 ^b^**	**0.017 ^b^**	**0.130**	**0.001 ^b^**
**Marital Status**	Single	52	75.32 (14.57)	66.03 (17.14)	70.67 (11.61)	58.46 (17.42)	75.00 (16.27)	80.77 (12.50)	78.08 (9.91)	52.88 (15.88)	42.69 (8.19)	72.69 (9.08)	63.33 (5.34)
	Married	734	72.48 (17.16)	64.51 (16.41)	68.49 (12.91)	57.83 (13.86)	73.16 (18.76)	80.49 (12.91)	78.39 (12.67)	50.80 (15.63)	44.96 (9.35)	74.60 (10.43)	63.59 (5.69)
	***p*** **-value**		**0.245**	**0.521**	**0.237**	**0.800**	**0.491**	**0.880**	**0.861**	**0.354**	**0.061**	**0.200**	**0.757**
**Nature of Job**	Sedentary	740	72.84 (16.97)	64.84 (16.41)	68.84 (12.79)	57.76 (14.19)	73.22 (18.73)	80.84 (12.65)	78.49 (12.52)	50.89 (15.69)	44.71 (9.13)	74.67 (10.10)	63.60 (5.58)
	Manual	46	69.93 (17.43)	60.87 (16.93)	65.40 (13.26)	59.78 (12.73)	74.35 (16.69)	75.22 (12.15)	76.52 (12.15)	51.74 (15.10)	46.52 (11.59)	71.30 (13.54)	63.00 (6.86)
	***p*** **-value**		**0.260**	**0.112**	**0.078**	**0.303**	**0.689**	**0.018 ^a^**	**0.301**	**0.722**	**0.301**	**0.104**	**0.489**
**Classification of Monthly Income^$^**	B40	732	72.43 (17.08)	64.34 (16.38)	68.39 (12.82)	57.88 (14.04)	73.25 (18.68)	80.38 (12.89)	78.39 (12.38)	51.17 (15.67)	44.78 (9.29)	74.37 (10.49)	63.57 (5.65)
	M40	52	75.00 (15.30)	68.91 (17.16)	71.96 (12.90)	57.69 (15.54)	73.46 (17.59)	81.54 (12.43)	77.69 (14.09)	48.08 (15.22)	45.48 (9.38)	75.77 (8.25)	63.56 (5.95)
	T20	2	100.00	50.00	75.00	60.00	80.00	100.00	90.00	40.00	35.00	76.67 (14.14)	64.17 (3.54)
	***p*** **-value**		**0.043 ^b^**	**0.070**	**0.120**	**0.973**	**0.875**	**0.083**	**0.390**	**0.237**	**0.274**	**0.614**	**0.989**
**Highest Education Level**	No formal education & Primary school	63	65.08 (22.14)	63.49 (16.90)	64.29 (15.07)	52.70 (10.96)	68.25 (20.20)	78.73 (12.38)	77.14 (10.69)	49.84 (14.54)	46.51 (10.19)	70.37 (11.72)	61.11 (5.41)
	Secondary school	373	71.18 (16.97)	63.72 (16.85)	67.45 (12.83)	58.71 (13.74)	73.57 (17.33)	79.89 (11.26)	78.55 (11.78)	53.14 (16.12)	45.07 (9.00)	73.78 (9.65)	63.93 (5.51)
	Certificate/diploma	165	74.44 (14.44)	65.96 (16.29)	70.20 (12.14)	58.36 (15.31)	72.73 (20.01)	80.73 (13.95)	78.06 (12.14)	49.21 (14.65)	46.67 (8.60)	75.19 (10.35)	63.46 (5.72)
	University degree and above	185	76.67 (16.04)	65.59 (15.60)	71.13 (12.01)	57.88 (14.11)	74.92 (19.06)	82.16 (14.88)	78.70 (14.69)	48.43 (15.40)	43.84 (10.10)	76.61 (10.75)	62.77 (5.83)
	***p*** **-value**		**≤0.001 ^b^**	**0.370**	**≤0.001 ^b^**	**0.017 ^b^**	**0.099**	**0.161**	**0.822**	**0.002 ^b^**	**0.217**	**≤0.001 ^b^**	**0.003 ^b^**
**Number of Lifetime Pregnancies (Parity)**	Nullparous	70	76.90 (13.98)	64.29 (16.37)	70.60 (10.87)	59.43 (14.74)	71.43 (17.55)	80.86 (13.38)	78.86 (11.23)	53.57 (14.25)	43.29 (9.12)	74.76 (9.42)	64.00 (5.06)
	1–3 Pregnancies	179	75.60 (14.70)	63.87 (16.71)	69.74 (12.27)	61.17 (13.67)	72.51 (19.19)	79.44 (13.14)	78.10 (13.18)	54.69 (16.05)	43.91 (8.37)	75.38 (9.72)	64.65 (5.48)
	More than 3 Pregnancies	155	75.81 (14.73)	65.59 (15.74)	70.70 (11.61)	58.00 (12.76)	73.03 (17.74)	78.71 (10.26)	78.06 (11.57)	53.61 (15.70)	44.19 (8.89)	73.81 (10.83)	63.57 (5.04)
	***p*** **-value**		**0.814**	**0.620**	**0.733**	**0.102**	**0.832**	**0.473**	**0.891**	**0.784**	**0.769**	**0.364**	**0.171**
**Current Menstrual Status**	Pre-menopause	99	76.94 (14.42)	63.64 (17.06)	70.29 (12.38)	62.02 (14.71)	74.39 (14.42)	79.39 (14.42)	79.39 (13.54)	56.97 (16.56)	43.94 (9.35)	73.33 (10.26)	64.97 (5.68)
	Peri-menopause	40	71.67 (17.78)	64.17 (13.37)	67.92 (12.60)	63.25 (14.57)	82.50 (11.27)	82.50 (11.27)	79.50 (13.19)	56.50 (16.58)	43.00 (8.83)	74.00 (10.97)	65.04 (5.13)
	Menopause	265	76.16 (14.02)	65.03 (16.39)	70.60 (11.40)	58.23 (12.77)	78.94 (12.19)	78.94 (11.37)	77.58 (11.56)	52.64 (14.92)	44.04 (8.43)	75.67 (10.11)	63.67 (5.06)
	***p*** **-value**		**0.138**	**0.756**	**0.407**	**0.012 ^b^**	**0.307**	**0.228**	**0.357**	**0.036 ^b^**	**0.781**	**0.242**	**0.055**

Low: 0–50%; Moderate 51–69%; High: 70% and above; a: indicates significant difference of *p* < 0.05, as assessed using independent t-test; b: indicates significant difference of *p* < 0.05, as assessed using post hoc analysis of ANOVA among the group in the same column. B40, subjects with household income <RM 7640; M40, subjects with household income RM 7640–15,159; T20, subjects with household income >RM 15,160.

**Table 4 ijerph-16-04115-t004:** Comparison of osteoprotective practices with sociodemographic factors.

Variable	Categories	N	n (%)												
			Dairy Intake		Calcium Supplement Intake		Coffee or Tea Intake		Alcohol Drinking		Smoking Status		Physical Activity Status		
			Do Not Drink	Regularly	No	Yes	Do Not Drink	Regularly	Non-Drinker	Ever-Drinker	Non-smoker	Ever-Smoker	Inactive	Minimally-Active	HEPA-Active
**Age Range (Years)**	40–59	457	281 (61.5)	176 (38.5)	399 (87.3)	58 (12.7)	90 (19.7)	367 (80.3)	406 (88.8)	51 (11.2)	360 (78.8)	97 (21.2)	239 (52.3)	163 (35.7)	55 (12.0)
	60 and above	329	211 (64.1)	118 (35.9)	269 (81.8)	60 (18.2)	56 (17.0)	273 (83.0)	284 (86.3)	45 (13.7)	254 (77.2)	75 (22.8)	140 (42.6)	150 (45.6)	39 (11.9)
	***p*-value**		**0.456**		**0.034 ***		**0.354**		**0.321**		**0.601**		**0.014 ***		
**Sex**	Men	382	272 (71.2)	110 (28.8)	341 (89.3)	41 (10.7)	51 (13.4)	331 (88.6)	304 (79.6)	78 (20.4)	223 (58.4)	159 (41.6)	168 (44.0)	159 (41.6)	55 (14.4)
	Women	404	220 (54.5)	184 (45.5)	327 (80.9)	77 (19.1)	95 (23.5)	309 (76.5)	386 (95.5)	18 (4.5)	391 (96.8)	13 (3.2)	211 (52.2)	154 (38.1)	39 (9.7)
	***p*-value**		**≤0.001 ***		**0.001 ***		**≤0.001 ***		**≤0.001 ***		**≤0.001 ***		**0.029 ***		
**Ethnicity**	Malay	342	198 (57.9)	144 (42.1)	303 (88.6)	39 (11.4)	59 (17.3)	283 (82.7)	324 (94.7)	18 (5.3)	240 (70.2)	102 (29.8)	218 (63.7)	96 (28.1)	28 (8.2)
	Chinese	363	251 (69.1)	112 (30.9)	298 (82.1)	65 (17.9)	68 (18.7)	295 (81.3)	291 (80.2)	72 (19.8)	299 (82.4)	64 (17.6)	126 (34.7)	177 (48.8)	60 (16.5)
	Indian and others	81	43 (53.1)	38 (46.9)	67 (82.7)	14 (17.3)	19 (23.5)	62 (76.5)	75 (92.6)	6 (7.4)	75 (92.6)	6 (7.4)	35 (43.2)	40 (49.4)	6 (7.4)
	***p*-value**		**0.001 ***		**0.045 ***		**0.432**		**≤0.001 ***		**≤0.001 ***		**≤0.001 ***		
**Marital Status**	Single	52	29 (55.8)	23 (44.2)	47 (90.4)	5 (84.6)	11 (21.2)	41 (78.8)	45 (86.5)	7 (13.5)	45 (86.5)	7 (13.5)	28 (53.8)	20 (38.5)	4 (7.7)
	Married	734	463 (63.1)	271 (36.9)	621 (9.6)	113 (15.4)	135 (18.4)	599 (81.6)	645 (87.9)	89 (12.1)	569 (77.5)	165 (22.5)	351 (47.8)	293 (39.9)	90 (12.3)
	***p*-value**		**0.302**		**0.319**		**0.583**		**0.826**		**0.164**		**0.538**		
**Nature of Job**	Sedentary	740	463 (62.6)	277 (37.4)	627 (84.7)	113 (15.3)	136 (18.4)	604 (81.6)	653 (88.2)	87 (11.8)	584 (78.9)	156 (21.2)	349 (47.2)	304 (41.1)	87 (11.8)
	Manual	46	29 (63.0)	17 (37)	41 (89.1)	5 (10.9)	10 (21.7)	36 (78.3)	37 (80.4)	9 (19.6)	30 (65.2)	16 (34.8)	30 (65.2)	9 (19.6)	7 (15.2)
	***p*-value**		**1.000**		**0.526**		**0.559**		**0.158**		**0.041 ***		**0.015 ***		
**Classification of Monthly Income^$^**	B40	732	463 (63.3)	269 (36.7)	623 (85.1)	109 (14.9)	134 (18.3)	598 (81.7)	642 (87.7)	90 (12.3)	573 (78.3)	159 (21.7)	353 (48.2)	293 (40.0)	86 (11.7)
	M40	52	28 (53.8)	24 (46.2)	43 (82.7)	9 (17.3)	11 (21.2)	41 (78.8)	46 (88.5)	6 (11.5)	39 (75.0)	13 (25.0)	24 (46.2)	20 (38.5)	8 (15.4)
	T20	2	1 (50.0)	1 (50.0)	2 (100)	-	1 (50.0)	1 (50.0)	2 (100)	-	2 (100)	-	2 (100)	-	-
	***p*-value**		**0.373**		**0.750**		**0.456**		**0.859**		**0.648**		**0.598**		
**Highest Education Level**	No formal education & Primary school	63	45 (71.4)	18 (28.6)	52 (82.5)	11 (17.5)	10 (15.9)	53 (84.1)	48 (76.2)	15 (23.8)	48 (76.2)	15 (23.8)	25 (39.7)	29 (46.0)	9 (14.3)
	Secondary school	373	234 (62.7)	139 (37.3)	315 (84.5)	58 (15.5)	76 (20.4)	297 (79.6)	329 (88.2)	44 (11.8)	287 (76.9)	86 (23.1)	177 (47.5)	147 (39.4)	49 (13.1)
	Certificate/diploma	165	108 (65.5)	57 (34.5)	146 (88.5)	19 (11.5)	26 (15.8)	139 (84.2)	148 (89.7)	17 (10.2)	134 (81.2)	31 (18.8)	78 (47.3)	68 (41.2)	19 (11.5)
	University degree and above	185	105 (56.8)	80 (43.2)	155 (83.8)	30 (16.2)	34 (18.4)	151 (81.6)	165 (89.2)	20 (10.8)	145 (78.4)	40 (21.6)	99 (53.5)	69 (37.3)	17 (9.2)
	***p*-value**		**0.146**		**0.537**		**0.578**		**0.031 ***		**0.713**		**0.539**		
**Number of Lifetime Pregnancies (Parity)**	Nullparous	70	39 (55.7)	31 (44.3)	61 (87.1)	9 (12.9)	12 (17.1)	58 (82.9)	64 (91.4)	6 (8.6)	65 (92.9)	5 (7.1)	37 (52.9)	29 (41.4)	4 (5.7)
	1-3 Pregnancies	179	96 (53.6)	83 (46.4)	139 (77.7)	40 (22.3)	44 (24.6)	135 (75.4)	173 (96.6)	6 (3.4)	174 (97.2)	5 (2.8)	91 (50.8)	67 (37.4)	21 (11.7)
	More than 3 Pregnancies	155	85 (54.8)	70 (45.2)	127 (81.9)	28 (18.1)	39 (25.2)	116 (74.8)	149 (96.1)	6 (3.9)	152 (98.1)	3 (1.9)	83 (53.5)	58 (37.4)	14 (9.0)
	***p*-value**		**0.950**		**0.212**		**0.382**		**0.181**		**0.112**		**0.671**		
**Current Menstrual Status**	Pre-menopause	99	50 (50.5)	49 (49.5)	85 (85.9)	14 (14.1)	20 (20.2)	79 (79.8)	97 (98.0)	2 (2.0)	95 (96.0)	4 (4.0)	61 (61.6)	28 (28.3)	10 (10.1)
	Peri-menopause	40	23 (57.5)	17 (42.5)	34 (85.0)	6 (15.0)	12 (30.0)	28 (70.0)	36 (90.0)	4 (10.0)	38 (95.0)	2 (5.0)	17 (42.5)	19 (47.5)	4 (10.0)
	Menopause	265	147 (55.5)	118 (44.5)	208 (78.5)	57 (21.5)	63 (23.8)	202 (76.2)	253 (95.5)	12 (4.5)	258 (97.4)	7 (2.6)	133 (50.2)	107 (40.4)	25 (9.4)
	***p*-value**		**0.643**		**0.222**		**0.461**		**0.118**		**0.636**		**0.169**		

*: indicates significant difference of *p* < 0.05, as assessed using chi-square among the group in the same column. B40, subjects with household income <RM 7640; M40, subjects with household income RM 7640–15,159; T20, subjects with household income >RM 15,160.

**Table 5 ijerph-16-04115-t005:** Correlation between osteoporosis knowledge, health beliefs, osteoprotective practices, and bone health status (N = 786).

Correlation		Overall (n = 786)														
		A	B	C	D	E	F	G	H	I	J	K	L	M	N	O
**A. Total Knowledge regarding Osteoporosis**	**r**	-	0.018	0.050	**0.169**	**0.151**	**−0.094**	**−0.145**	**0.219**	**0.108**	**0.086**	0.067	−0.014	**−0.083**	−0.010	0.027
	***p***		0.605	0.160	**≤0.001**	**≤0.001**	**0.008**	**≤0.001**	**≤0.001**	**0.002**	**0.016**	0.060	0.686	**0.019**	0.773	0.456
**B. Perceived Susceptibility to Osteoporosis**	**r**		-	**0.196**	0.033	0.032	**0.165**	0.006	**0.074**	0.034	**0.078**	−0.028	−0.037	−0.056	−0.002	−0.002
	***p***			**≤0.001**	0.362	0.365	**≤0.001**	0.864	**0.038**	0.335	**0.028**	0.435	0.294	0.120	0.954	0.964
**C. Perceived Seriousness of Osteoporosis**	**r**			-	0.040	0.047	0.051	0.006	0.008	0.036	−0.021	0.028	0.022	0.022	0.001	0.019
	***p***				0.265	0.187	0.156	0.870	0.824	0.313	0.565	0.433	0.539	0.530	0.988	0.596
**D. Perceived Benefits of Exercise**	**r**				-	0.173	**–0.110**	−0.050	**0.183**	0.043	0.017	0.014	0.046	−0.021	**0.110**	−0.067
	***p***					**≤0.001**	**0.002**	0.159	**≤0.001**	0.229	0.642	0.699	0.201	0.558	**0.002**	0.060
**E. Perceived Benefits of Calcium Intake**	**r**					-	0.077	0.006	0.147	**0.071**	−0.002	0.027	−0.032	0.069	−0.039	−0.073
	***p***						0.855	0.865	**≤0.001**	**0.046**	0.950	0.454	0.367	0.053	0.272	0.079
**F. Barriers to Exercise**	**r**						-	**0.127**	**–0.117**	−0.068	−0.025	−0.028	**–0.104**	−0.024	**–0.205**	**–0.063**
	***p***							**≤0.001**	**≤0.001**	0.055	0.479	0.437	**0.003**	0.502	**≤0.001**	**0.041**
**G. Barriers to Calcium Intake**	**r**							-	−0.054	**–0.080**	−0.026	−0.013	−0.013	**0.094**	−0.036	−0.010
	***p***								0.129	**0.024**	0.469	0.709	0.711	**0.009**	0.316	0.778
**H. Health Motivation**	**r**								-	**0.164**	**0.073**	−0.017	−0.056	**−0.072**	**0.120**	−0.038
	***p***									**≤0.001**	**0.040**	0.632	0.117	**0.044**	**0.001**	0.289
**I. Dairy Intake**	**r**									-	0.043	−0.030	0.039	**−0.129**	**0.102**	0.036
	***p***										0.227	0.406	0.270	**≤0.001**	**0.004**	0.308
**J. Calcium Supplement Intake**	**r**										-	−0.047	−0.026	**−0.076**	0.056	**0.082**
	***p***											0.192	0.463	**0.033**	0.117	**0.021**
**K. Coffee or Tea Intake**	**r**											-	**0.118**	**0.158**	0.005	−0.059
	***p***												**0.001**	**≤0.001**	0.887	0.100
**L. Alcohol Drinking**	**r**												-	**0.254**	**0.107**	0.004
	***p***													**≤0.001**	**0.003**	0.902
**M. Smoking Status**	**r**													-	0.006	**−0.079**
	***p***														0.864	**0.026**
**N. Physical Activity Status**	**r**														-	−0.041
	***p***															0.253
**O. Bone Health Status**	**r**															-
	***p***															

Number in bold indicated significant association between the compared variables.

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
