# Peer review of "Levels of Knowledge, Beliefs, and Practices Regarding Osteoporosis and the Associations with Bone Mineral Density among Populations More Than 40 Years Old in Malaysia"

_ijerph, 2019, doi:10.3390/ijerph16214115_

Round 1
Reviewer 1 Report
The manuscript is interesting and well thought-out.
The aim of the study is to bridge the research gap by determining the levels of knowledge, beliefs, practices regarding osteoporosis and their association with BMD in Malaysia.
However I would like the authors to:
Add some hints about the pathogenesis of osteoporosis and the multifactorial role of inflammation and emerging therapies Ciccarelli et al. Glucocorticoids in Patients with Rheumatic Diseases: Friends or Enemies of Bone? Curr Med Chem. 2015; 22(5):596-603) Ginaldi et al. Increased levels of interleukin 31 (IL-31) in osteoporosis. BMC Immunol. 2015 Oct 8;16:60) Ginaldi et al. Interleukin-33 serum levels in postmenopausal women with osteoporosis. Sci Rep. 2019; doi: 10.1038/s41598-019-40212-6 De Martinis M et al. Osteoporosis: Current and emerging therapies targeted to immunological checkpoints. Curr Med Chem. 2019 Jul 30. doi: 10.2174/0929867326666190730113123.Please specify inclusion and exclusion criteria
Line 89: “They a questionnaire…..” ? Please revise
Reference numbers: reference 22 revise
Author Response
Dear reviewer,
Thank you for reviewing our manuscript. We appreciated your constructive comments and they are responded according in the file attached.
Regards,
Chan Chin Yi

Reviewer 2 Report
General comments:
In this manuscript, Chan et al build upon their earlier work (referred to here as their “pilot study”) to investigate the level of knowledge related to bone health / osteoporosis risk factors among different age groups and other demographic variables for a population in Malaysia.
In general, the paper is very clearly written and presents statistically significant results. The findings from this study build upon the authors’ earlier work and extend the impact of their research into a broader population base. I have only a few suggestions for improvement and clarification.
Specific comments:
Introduction: As this study was conducted in an Asian population, and those of Asian ethnicity are at elevated risk for osteoporosis as compared to other ethnic groups, I believe this topic is important to note in the introduction. I recommend adding a few sentences to the introduction on the demographics of those at greatest risk for osteoporosis, and using that as another line of justification for why the studies presented in this manuscript are of value to the greater research community. Methods: In the “Subjects” section, the authors include a categorization of study subjects based on household income. While this is certainly of relevance to the study as presented, they should provide additional explanation here for why this variable would be of interest. For example, does income correlate with health status in this study population? Or is there any evidence from past studies that income would correlate with health knowledge? This would be relevant to mention here. Methods: In the “statistics” section, the authors should provide some sort of description of how sample sizes were chosen – i.e., can the reader be confident that an adequate sample size was included for proper statistical power? This is relevant, because the authors refer to their previous “pilot study” in in the introduction – presumably this current study is meant to provide a definitive answer to their research question, and therefore it is critical to assure the reader that it was adequately powered to provide statistically significant results if possible. Results, section 3.2: “…yellow dhal” is not a commonly recognized food globally; the authors may consider explaining what this is in parentheses next to it (lentils?)
Author Response

(The authors gave the same response as above.)
